# Protective activities of distinct omega-3 enriched oils are linked to their ability to upregulate specialized pro-resolving mediators

Agua Sobrino[1], Mary E. Walker[1], Romain A. Colas[1], Jesmond Dalli[1,2]*

1 William Harvey Research Institute, Barts and The London School of Medicine and Dentistry, Queen Mary University of London, London, United Kingdom, 2 Centre for Inflammation and Therapeutic Innovation, Queen Mary University of London, London, United Kingdom

* j.dalli@qmul.ac.uk

## Abstract

Clinical studies using a range of omega-3 supplements have yielded conflicting results on their efficacy to control inflammation. Omega-3 fatty acids are substrate for the formation of potent immune-protective mediators, termed as specialized pro-resolving mediators (SPM). Herein, we investigated whether observed differences in the potencies of distinct omega-3 supplements were linked with their ability to upregulate SPM formation. Using lipid mediator profiling we found that four commercially available supplements conferred a unique SPM signature profile to human macrophages, with the overall increases in SPM concentrations being different between the four supplements. These increases in SPM concentrations were linked with an upregulation of macrophage phagocytosis and a decreased uptake of oxidized low-density lipoproteins. Pharmacological inhibition of two key SPM biosynthetic enzymes 5-Lipoxygenase or 15-Lipoxygenase reversed the macrophage-directed actions of each of the omega-3 supplements. Furthermore, administration of the two supplements that most potently upregulated macrophage SPM formation and reprogrammed their responses *in vitro*, to APOE$^{-/-}$ mice fed a western diet, increased plasma SPM concentrations and reduced vascular inflammation. Together these findings support the utility of SPM as potential prognostic markers in determining the utility of a given supplement to regulate macrophage responses and inflammation.

## Introduction

Cardiovascular diseases (CVD) represent the most frequent causes of death worldwide. Most of these are associated with the development of atherosclerosis, which is characterized by chronic unresolved inflammation leading to plaque formation and progression [1–3]. In addition to anti-inflammatory therapies, recent approaches in limiting exuberant inflammatory responses characteristic of chronic inflammatory conditions include immunonutrition [4–6]. In this context several studies have highlighted the utility of omega-3 enriched supplements in the prevention of CVD [6–8]. The American Heart Association science advisory committee recommends the administration of marine-based omega-3 oils for patients with prevalent coronary heart disease [9, 10]. Recent clinical studies have further underscored the utility of omega-3 fatty acids in the treatment of CVD. Whereby, in the REDUCE-IT trial Icosapent

**Data Availability Statement:** All relevant data are within the manuscript and its Supporting Information files.

**Funding:** This work was supported by funding from the European Research Council (ERC) under the European Union's Horizon 2020 research and innovation programme (grant no: 677542) and the Barts Charity (grant no: MGU0343) to JD. JD is also supported by a Sir Henry Dale Fellowship jointly funded by the Wellcome Trust; the Royal Society (grant 107613/Z/15/Z); and a grant from Standard Process Inc. The funders had no role in study design, data collection and analysis, decision to publish, or preparation of the manuscript.

**Competing interests:** The authors have read the journal's policy and have the following competing interests: This study was partially funded by Standard Process Inc. in the form of a grant awarded to JD. Standard Process Inc. has three marketed products associated with this research, namely, Standard Process Olprima DHA, Olprima EPA and Olprima EPA/DHA. The company is not planning to develop new products using these ingredients in the near future, and there are no patents envisioned at this moment. The company has not pursued any commercial opportunities with either algal or PRM 300 oil. This does not alter our adherence to PLOS ONE policies on sharing data and materials. There are no further patents, products in development or marketed products associated with this research to declare.

ethyl (IPE), a highly purified form of eicosapentaenoic acid (EPA) ethyl ester, added to a statin reduced initial cardiovascular (CV) events by 25% and total CV events by 32 [11]. The recently concluded EVAPORATE study demonstrated that administration of IPE together with statin led to increased regression of low-attenuation plaque volume on multidetector computed tomography compared with placebo over 18 months [12] Furthermore, meta-analysis and meta-regression of interventional trials provides additional support for the use of enriched omega-3 FAs, EPA and docosahexaenoic acid (DHA) as an effective strategy for CVD prevention [13]. Of note, the utility of omega-3 supplements does not appear to be universal, with several studies suggesting that some omega-3 supplements display limited benefit in reducing inflammation [14–16]. Several aspects have been suggested for such apparently discordant findings, including heterogeneity in study populations, sample size, interventions and outcomes. Among the aspects that have to date been relatively understudied are the impact of supplement origin (i.e. algal, fish oil or plant-derived) as well as formulation (e.g. ethyl ester *versus* triglyceride form). This aspect has been particularly hampered by the limited availability of robust biomarkers that can be used to determine the ability of a specific supplement in regulating immune responses. In this context, recent studies have suggested that distinct FA pools such as red blood cells and another plasma FA pools may be useful to evaluate DHA and EPA loading after fatty acid supplementation [17, 18].

It is now well appreciated that the termination of acute inflammation is a coordinated response. Studies investigating the molecular mechanism promoting the termination of acute inflammation uncovered a central role for omega-3 derived mediators, termed as specialized pro-resolving mediators (SPM), in reprogramming immune responses, regulating leukocyte trafficking and counter regulating the production of inflammatory mediators [19, 20]. Biochemical experiments addressing the formation of these potent autacoids demonstrate that SPM are produced *via* the stereoselective conversion of omega-3 FA, including EPA and DHA [19, 20]. SPM display potent biological actions in reprogramming macrophage responses, upregulating their ability to clear apoptotic cells and bacteria as well as to counter regulate the production of inflammatory mediators[19, 20]. Recent results demonstrate that SPM are dysregulated in patients with cardiovascular disease, these include a reduction in plasma concentrations of n-3 docosapentaenoic acid-derived resolvins ($RvD_{n-3\ DPA}$) as well as a decrease in aortic tissues concentrations of the DHA-derived resolvin (Rv)D1 [21, 22]. Furthermore, administration of $RvD5_{n-3\ DPA}$ or the EPA-derived RvE1 protect against atherosclerotic plaque formation [21, 23], whereas the DHA-derived RvD1 promotes atherosclerotic plaque regression[17]. These finding suggest that restoring SPM concentrations may represent a promising alternative to current therapies to promote the termination of vascular inflammation.

Thus, in the present study we aimed at addressing the utility of SPM pathways as biomarkers in determining the immune regulatory potential of different omega-3 supplements. For this purpose, we evaluated the ability of four different commercially available omega-3 enriched supplements to regulate key immune responses in the resolution of atherosclerotic inflammation *in vitro* and vascular inflammation *in vivo*. We also related these findings with the ability of these supplements to upregulate SPM formation by macrophages *in vitro* and plasma SPM concentrations *in vivo*.

## Results

### Characteristic concentrations of FA substrates and SPM precursors in distinct omega-3 supplements

In order to establish whether the four commercial supplements could differentially impact SPM production we first evaluated the concentrations for SPM substrates (arachidonic acid

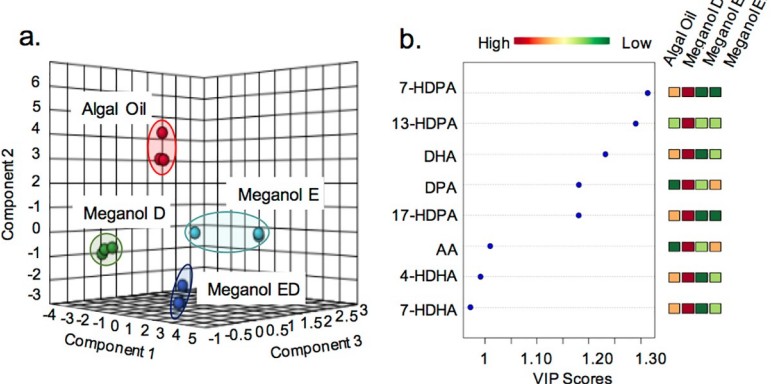

**Fig 1. FA and SPM precursors concentrations are differentially present in the four selected enriched oils studied.**
Substrate and precursor concentrations were assessed using LC-MS/MS based profiling. The relationship between the
concentrations of the molecules in each of the supplements tested was evaluated using Partial Least Square
Discriminant Analysis (PLS-DA). (**a**) Score Plots (**b**) Variable importance in projection (VIP) scores of the eight SPM
precursors/substrates with the greatest differences between the four groups. Results are representative of 3
determinations.

(AA), DHA, n-3 DPA and EPA) and precursors (18-HEPE, 17-HDHA, 7-HDHA, 4-HDHA,
17-HDPA, 13-HDPA, 7-HDPA, 15-HETE and 5-HETE) in these supplements. We then
employed multivariate analysis to evaluate whether the overall profiles for these molecules
were similar or different. Here we found that each of the oils presented a distinct profile as
demonstrated by a separation of the clusters representing each of the oils in PLS-DA (Fig 1A
and Table 1). Assessment of VIP scores for the identified FA and SPM precursors indicate that
amongst the precursors that are differentially expressed between the four supplements,
7-HDPA and 13-HDPA were the ones with greater differences between the four supplements
(Fig 1B). Of note, we also observed that Meganol D displayed the highest concentrations in
omega-3 FA and SPM precursors (Fig 1B).

**Table 1. Omega-3 enriched oil FA and SPM precursor concentrations.**

| | Algal Oil | | | Meganol D | | | Meganol E | | | Meganol ED | | |
|---|---|---|---|---|---|---|---|---|---|---|---|---|
| **FA & SPM precursors** | Mean | | SEM | Mean | | SEM | Mean | | SEM | Mean | | SEM |
| 17-HDHA | 4,466.2 | ± | 472.9 | 623.5 | ± | 40.5 | 0.0 | ± | 0.0 | 295.9 | ± | 18.0 |
| 7-HDHA | 240.6 | ± | 23.7 | 354.4 | ± | 36.4 | 0.0 | ± | 0.0 | 35.2 | ± | 43.2 |
| 4-HDHA | 484.5 | ± | 61.9 | 1378.6 | ± | 183.3 | 0.0 | ± | 0.0 | 397.2 | ± | 21.9 |
| **DHA** | 28,361.2 | ± | 4800.4 | 364,805.8 | ± | 89,365.3 | 965.2 | ± | 365.5 | 24,638.6 | ± | 12594.2 |
| 17-HDPA | 77.8 | ± | 13.2 | 423.9 | ± | 31.7 | 0.0 | ± | 0.0 | 0.0 | ± | 0.0 |
| 13-HDPA | 0.0 | ± | 0.0 | 39.0 | ± | 10.6 | 0.0 | ± | 0.0 | 0.0 | ± | 0.0 |
| 7-HDPA | 5.6 | ± | 3.8 | 136.3 | ± | 20.7 | 0.0 | ± | 0.0 | 0.0 | ± | 0.0 |
| **DPA** | 244.5 | ± | 45.9 | 101,079.4 | ± | 28,463.2 | 800.0 | ± | 359.9 | 2,381.8 | ± | 976.5 |
| 18-HEPE | 30.0 | ± | 4.5 | 1782.6 | ± | 226.2 | 40,88.9 | ± | 820.8 | 1,473.3 | ± | 35.7 |
| **EPA** | 1,235.0 | ± | 189.1 | 468,964.9 | ± | 91,930.3 | 36,357.6 | ± | 17,485.4 | 48,749.9 | ± | 13766.1 |
| 15-HETE | 26.1 | ± | 3.2 | 95.4 | ± | 5.2 | 75.5 | ± | 9.7 | 15.9 | ± | 19.5 |
| 5-HETE | 54.1 | ± | 33.3 | 301.8 | ± | 64.1 | 486.0 | ± | 90.7 | 205.1 | ± | 22.1 |
| **AA** | 190.3 | ± | 40.0 | 51,388.6 | ± | 14,747.3 | 2,329.5 | ± | 1,583.1 | 1,870.6 | ± | 700.2 |

Results are mean of 3 replicates. Concentration is expressed as pg / 25 μL of oil.

## Omega-3 enriched oils regulate distinct macrophage phenotypic markers

Although many cells are involved in the development and progression of atherosclerosis, macrophages are fundamental contributors, and understanding the phenotypic heterogeneity of macrophages and functional consequences contribute to determine their potential role in lesion development [24]. Thus, we next tested whether the four oils regulated monocyte-derived macrophage responses in the presence of an inflammatory phenotype. For this purpose, we incubated monocyte-derived macrophages with lipopolysaccharide (LPS), a prototypical inflammatory agonist, and treated these cells with or without one of the four test oils. Since we found that different oils contained distinct concentrations of both FA substrates and SPM precursors (Fig 1), and in order to better evaluate the impact of each oil on pro-resolving pathways we incubated the cells with 200 pg of the combined concentrations of SPM precursors 18-HEPE, 17-HDHA, 7-HDHA, 4-HDHA, 17-HDPA, 13-HDPA, 7-HDPA, 15-HETE, 5-HETE and their parent fatty acids. This concentration was selected since it is relevant to plasma levels of these molecules achieved in humans after supplementation with fish oils [21]. Evaluation of the expression of a panel of phenotypic markers using flow cytometry demonstrated that all the oils tested differentially macrophage phenotype as demonstrated by a shift in the cell clusters representing cells incubated with the distinct omega-3 enriched oils (Fig 2A and S1 Table in S1 File).

To better evaluate the impact of these oils on regulating LPS mediated responses we compared the expression of phenotypic markers on LPS treated cells with that found on monocyte-derived macrophages that were not incubated with LPS (M0 macrophages). This analysis demonstrated that Algal oil co-incubation led to a further increase in CD14 expression when compared to both cells incubated with LPS alone and M0 macrophages. LPS incubation was found to decrease CD36 expression, an observation that was reversed when cells were incubated with Algal oil, Meganol E or Meganol ED. Intriguingly, we found that LPS marginally increased the expression of all the other markers investigated. Co-incubation of cells with Meganol D further increased CD68, MerTK and CD300f expression. Incubation of cells with Meganol E increase CD68 and CD14 expression when compared with cells incubated with LPS alone. Whereas incubation of cells with Meganol ED upregulated CD14, CD300f and CD163 expression (S1 Table in S1 File). Together these findings indicate that each of the oils leads to a differential regulation of macrophage responses to LPS stimulation.

## Omega-3 enriched oils differentially regulate SPM production in human monocyte-derived macrophages

SPM play a central role in orchestrating the resolution of inflammation and human macrophages are one of the major cellular sources of these mediators in the innate immune system [25, 26]. Thus, having observed that the distinct oils tested regulated macrophage phenotype we next queried whether this was linked with changes in SPM production. To test this, using LC-MS/MS-based methodologies we assessed the SPM profiles of human monocyte-derived macrophages incubated with equal concentrations of the test oils and compared these to those obtained from macrophages incubated in media with vehicle alone. Partial least squared discriminant analysis (PLS-DA) of lipid mediator profiles from these incubations demonstrated that each of the oils led to a distinct shift in lipid mediator profiles as demonstrated by a separation of the clusters representing macrophages incubated with the different oils (Fig 3A and S2 Table in S1 File). Assessment of VIP scores demonstrated that among the top mediators to be differentially produced in these incubations were mediators from the DHA and n-3 DPA bioactive metabolomes, primarily those produced *via* the interaction of 5-lipoxygenase (ALOX5) and ALOX15. These included DHA-derived RvD1, RvD2 and the n-3 DPA-derived RvT1 and RvD1$_{n-3 \text{ DPA}}$ (Fig 3B). Of note, assessment of the concentrations of mediators that displayed the highest degree of differential

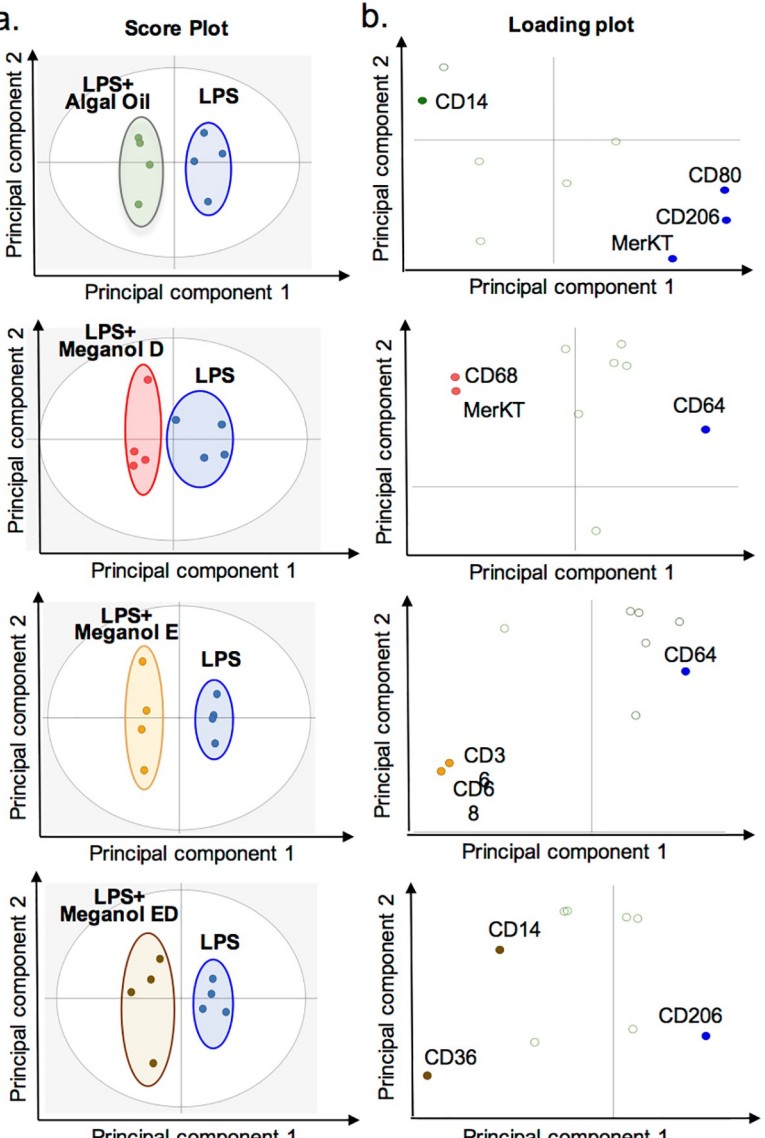

**Fig 2. Differential regulation of macrophage phenotypic markers by distinct omega-3 enriched oils.** Human monocyte derived macrophages were incubated with LPS (1 ng/ mL, 24 h) and with the indicated oils (200 pg of SPM precursors, 24 h; *see methods for details*) or vehicle (PBS + 0.1% EtOH + 1 ng/ mL, 24 h). The expression of phenotypic markers was assessed using flow cytometry and differential expression was assessed using orthogonal partial least squares discriminant analysis (OPLS-DA). Variable Importance in Projection (VIP) score greater than 1. (**a**) Score Plots (**b**) Loading Plots. Results are representative of n = 4 donors.

regulation (i.e., highest VIP scores) between the four oils demonstrated that their concentrations were highest in cells supplemented with Meganol D (Fig 3B and S2 Table in S1 File).

## Dose-dependent increases in macrophage phagocytosis of S. aureus bioparticles by omega-3 enriched oils

Phagocytosis, an essential mechanism in the termination of acute inflammation, is considered to be a key pro-resolving action exerted by SPM [27]. Thus, to gain further insights into the pro-resolving properties of the different oil preparations we assessed their ability to upregulate

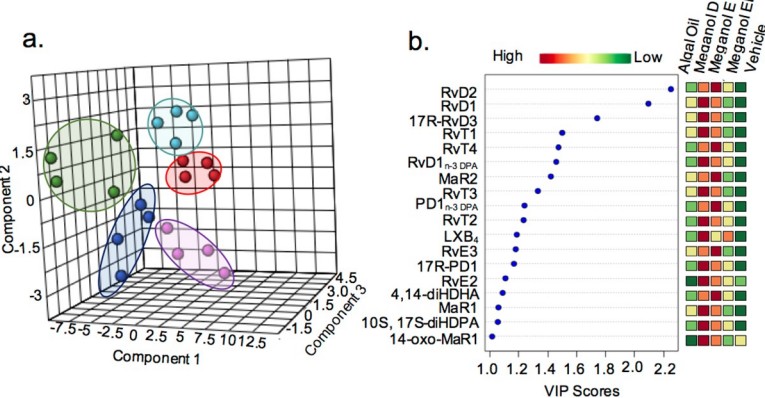

**Fig 3. Omega-3 enriched oils differentially regulate SPM production in human monocyte-derived macrophages.**
Human monocyte-derived macrophages were incubated with LPS (1 ng/ mL, 24 h) and with the indicated oils (200 ng of SPM precursors, 24 h) or vehicle (PBS + 0.1% EtOH). Lipid mediator profiles were assessed using LC-MS/MS-based profiling and differential expression was assessed using partial least squares discriminant analysis (PLS-DA). (**a**) Score Plots. Red dots correspond to vehicle, green dots to Algal Oil, dark blue dots to Meganol D Oil, light blue dots to Meganol E and pink dots to Meganol ED (**b**) Variable importance in projection (VIP) scores of 18 lipid mediators with the greatest differences in concentrations between the five groups. Results are representative of n = 4 donors.

macrophage phagocytosis. Here we found that all four oils tested increased the phagocytosis of *S. aureus* bioparticles in a dose-dependent manner with statistically significant increases observed at concentrations as low as 100 pg/ mL for Meganol D (Fig 4). At 120 pg/ mL we found that Algal oil and Meganol D were the most potent at upregulating macrophage phago-cytosis up to ~9% and ~6% of increase respectively. Highest potency of ~13% increase was reached by Meganol D at a maximum concentration of 200 pg of precursor concentration (Fig 4). These results demonstrate that each of the oils is able to engage pro-resolving responses in human monocyte-derived macrophages.

## Omega-3 enriched oils reduce neutral lipids uptake by monocyte-derived macrophages

Macrophage uptake and intracellular accumulation of neutral lipids leads to foam cell forma-tion, a phenomenon that is linked with the perpetuation of vascular inflammation in athero-sclerosis [28–30]. Recent studies demonstrate that SPM display protective actions in preventing and limiting vascular inflammation and neutral lipid accumulation in the vascular wall [31]. Thus, we next tested the ability of each of the oils to regulate neutral lipid load in foam cells *in vitro*. For this purpose, we determined the ability of these oils to regulate the uptake of fluorescently labelled oxLDL (Fig 5). Here, we found a dose-dependent reduction in neutral lipid load in macrophages incubated with all four oils, with a decrease in neutral lipid accumulation reaching statistical significance in cells incubated with Algal oil, Meganol D and Meganol ED. Of note, Meganol ED displayed the highest potency at limiting neutral lipid accumulation in monocyte-derived macrophages where we observed ~13% decrease in neutral lipid content at the lowest dose of the oil tested (Fig 5).

## Inhibition of SPM formation partially abolishes the pro-resolving actions of omega-3 enriched oils on human monocyte-derived macrophage

We next queried whether the observed increased in SPM following Omega-3 enrich oil addi-tion was linked with the upregulation of pro-resolving responses in human monocyte-derived

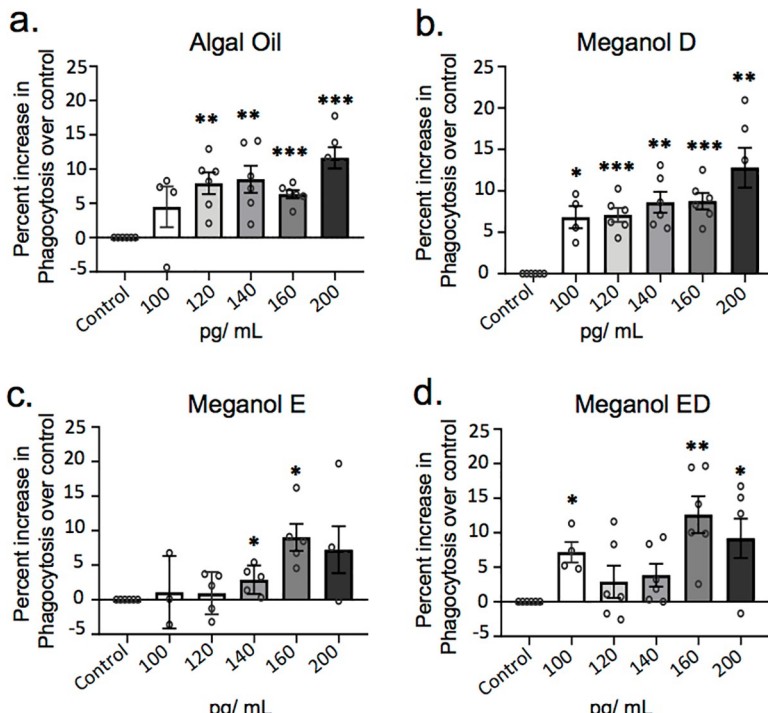

**Fig 4. Omega-3 enriched oils dose-dependently increase macrophage phagocytosis of *S. aureus* bioparticles.**
Human monocyte-derived macrophages were incubated for 24 h with the indicated concentrations of oils (pg/ mL of
SPM precursors) or vehicle (PBS + 0.1% EtOH) then fluorescent bacterial bioparticles were added and phagocytosis
was assessed after 1h of incubation. Results are expressed as percent increase over control for (**a**) Algal oil (**b**) Meganol
D (**c**) Meganol E and (**d**) Meganol ED. X-axis represents oil concentration SPM precursors (pg/ mL). Results
are ± SEM. n = 6 donors. *p< 0.05; **p< 0.01; ***p< 0.001 using one-sample t-test.

macrophages. For this purpose, we used a pharmacological approach where macrophages were
incubated with an ALOX5 or ALOX15 inhibitor, given that mediators produced by these
enzymes were amongst the ones that were upregulated to the greatest extent (Fig 3). Here we
found that incubation of macrophages with inhibitors to ALOX5 reversed the ability of each of
the oils to upregulate macrophage phagocytosis (p< 0.05) (Fig 6A). Similarly, inhibition of
ALOX15 activity reversed the ability of each of the oils tested to upregulate macrophage
phagocytosis, an observation that reached statistical significance for cells incubated with Algal
Oil and Meganol D (p< 0.05; Fig 6B). These results support a role for SPM formation in medi-
ating the ability of omega-3 enriched oils to upregulate macrophages phagocytosis.

## Algal oil and Meganol D reduce early atherosclerotic lesions in Apoe$^{-/-}$ mice and upregulate SPM formation

In order to test whether the protective actions of omega-3 enriched oils observed *in vitro* are
also retained *in vivo*, we next tested their ability to protect against vascular disease. For this
purpose, we limited our analysis to Algal oil and Meganol D given that these two oils were
found to be amongst the most potent at regulating macrophage biological actions, cells that
are central in the pathophysiology of vascular diseases [32]. To evaluate whether these two oils
protected against early atherosclerotic lesion formation we fed male and female *Apoe$^{-/-}$* mice
with a western diet for 5 weeks and then treated them with one of the oils, which were admin-
istered as 9.2 µg of the SPM precursors per mouse per day, or vehicle for a 2-week period. We
then assessed vascular lesion size using oil-red O staining. Results from these experiments

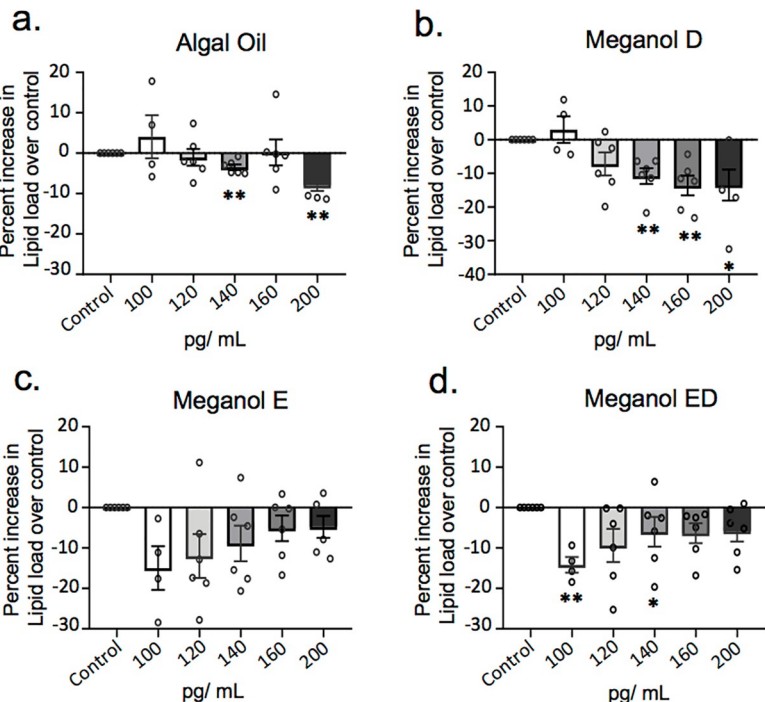

**Fig 5. Omega-3 enriched oils dose-dependently decrease macrophage lipid uptake.** Human monocyte-derived macrophages were incubated with the indicated concentrations of oil (pg/ mL of SPM precursors) or vehicle control (PBS + 0.1% EtOH) and incubated for 24 h with fluorescently labelled oxidized LDL. Lipid load was determined using a fluorescent plate reader. Results are expressed as percent increase over control for (**a**) Algal oil (**b**) Meganol D (**c**) Meganol E and (**d**) Meganol ED. Y-axis represents oil concentration SPM precursors (pg/ mL). Results are ± SEM. n = 6 donors. *p< 0.05; **p< 0.01 using one-sample t-test.

demonstrated that each of the oils reduced vascular lesion size, which reached statistical significance for mice treated with Meganol D (Fig 7A and 7B).

To determine whether the protective actions exerted by both oils were linked with changes in peripheral blood SPM concentrations we next assessed plasma SPM profiles in these mice

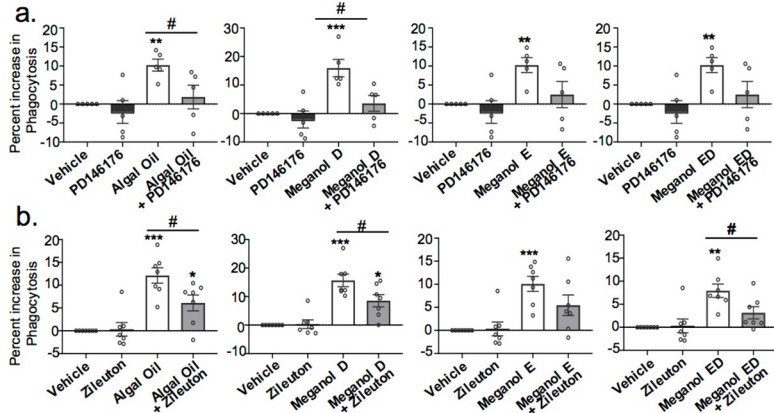

**Fig 6. Inhibition of ALOX5 and ALOX15 reduced the ability of omega-3 enriched oil to upregulate to macrophage phagocytosis of *S. aureus* bioparticles.** Human monocyte-derived macrophages were incubated for 24 h with each oil (200 pg/ mL of SPM precursors) or vehicle (PBS + 0.1% EtOH) in combination or not with (**a**) ALOX5 inhibitor (10 μM Zileuton) or (**b**) ALOX15 inhibitor (25 μM PD146176). Fluorescent bacterial bioparticles were added and phagocytosis was assessed after 1h of incubation. Results are expressed as percent increase over control. Results are ± SEM. n = 5–7 donors. *p < 0.05; **p< 0.01; ***p< 0.001 one-sample t-test. # p< 0.05 using Mann Whitney test.

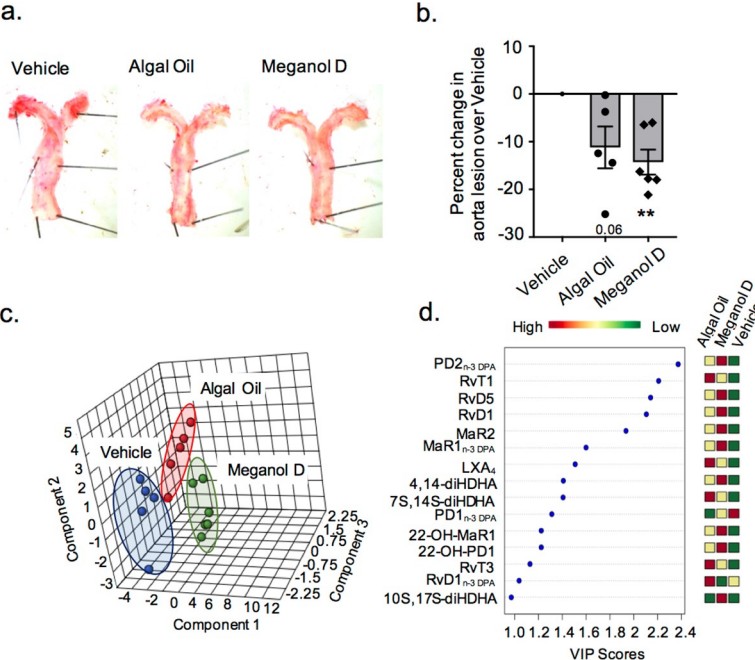

**Fig 7. Algal oil and Meganol D decreased early vascular lesions and upregulate plasma SPM in APOE$^{-/-}$ mice.**
Mice were fed a western diet for 5 weeks, then they were administered Algal oil or Meganol D at dose of 9.2 μg of SPM precursors per mouse/day or vehicle (0.1% EtOH) for a 2-week period. (**a**) Representative images of aortas from mice fed western diet and treated with Meganol D Algal Oil, or vehicle. (**b**) Aortas were harvested and lipid content determined using oil-red O staining. Results are representative of n = 5–6. $^{**}$p< 0.01 using one-sample t-test. (**c-d**) Plasma was harvested and lipid mediator profiles were assessed using LC-MS/MS based profiling. Differential expression was determined using partial least squares discriminant analysis (PLS-DA) (**c**) Score Plots (**d**) Variable importance in projection (VIP) scores of 15 lipid mediators with the greatest differences in concentrations between the three groups. Results are representative of n = 5–6 mice per group.

using lipid mediator profiling. Here we found that each of the oils lead to distinct plasma lipid mediator profiles as demonstrated by a separation of the clusters representing mice treated with either of the enriched oils or vehicle in PLS-DA (Fig 7C and S3 Table in S1 File). Assessment of VIP scores for the identified mediators demonstrated that Meganol D supplementation increased plasma levels of several DHA and n-3 DPA-derived SPM including PD2$_{n-3\ DPA}$, RvD1, RvD5, MaR2 or MaR1$_{n-3\ DPA}$ (Fig 7D and S3 Table in S1 File). Whereas, Algal oil increased the plasma concentrations of n-3 DPA- derived RvT1 and RvT3, the vasculoprotective RvD1$_{n-3\ DPA}$ [33] and the endothelial-protective AA-derived LXA$_4$ [6, 9, 10] (Fig 7D and S3 Table in S1 File). Together, these findings suggest that *via* upregulation of peripheral blood SPM concentrations the two omega-3 enriched oils exert vasculoprotective actions.

## Discussion

In the present study, we sought to establish the potential mechanisms underpinning the differential activity of distinct omega-3 oils and the potential utility of SPM as biomarkers for determining the immune-regulatory activity of such oils. Our findings demonstrate that different enriched oils contain distinct amounts of SPM precursors and substrates. Even after normalizing for SPM precursor levels we found that each displayed a unique bioactive profile in their abilities to regulate macrophage responses *in vitro* and vascular inflammation *in vivo*. Notably, differential activity of these oils was linked with their ability to upregulate SPM biosynthesis both *in vitro* and *in vivo*. Together, these findings indicate that differences in vascular

protection from distinct omega-3 enriched oils may be, at least in part, due to their ability to upregulate SPM formation in target cells and organs.

While it is generally appreciated that omega-3 supplements exert protective actions in the regulation of CVD[7, 14–16, 34–37], there does not appear to be a consensus on the matter given that a number of studies failed to demonstrate a benefit when patients were treated with these supplements [23]. In the present study, we found that conversion of omega-3 fatty acids and SPM precursors to bioactive mediators is not equivalent between different supplements even when the amounts of these precursors are normalised. This suggests that the forms in which these precursors are incorporated into these supplements together with other components present in such formulations may impact their suitability for conversion to SPM. This aspect is of relevance since we found that those supplements that increased SPM formation to the greatest extent, Meganol D and Algal oil, were also the ones that most potently regulated human macrophage responses. Furthermore, pharmacological inhibition of SPM formation reversed the ability of these supplements to regulate human macrophages.

Macrophage phenotypic heterogeneity contributes to determine their potential role in atherosclerotic lesion development [38]. In the present study, we found that all four oils differentially regulated the expression of macrophage phenotypic markers in response to the pro-inflammatory agonist LPS (Fig 2). Of interest, Meganol D, which was one of the most potent at regulating macrophage responses *in vitro* and reducing vascular inflammation *in vivo* and upregulated the expression of CD68 and c-mer tyrosine kinase (MerTK) receptor expression on monocyte-derived macrophages in vitro (Fig 2). It is well established that MerTK receptor promotes resolution of inflammation, for example, it is described to be upregulated in phagocytosis of apoptotic cells in atherosclerotic lesions, and is linked to RvD1's ability to enhance lesional macrophage efferocytosis [39]. In addition, MerTK signalling is also linked with regulating the subcellular localization of ALOX5 and the biosynthesis of $LXA_4$ in macrophages [40]. This is in line with findings presented herein, where we observed higher concentration of $LXA_4$ together with other ALOX5-derived SPM in macrophages treated with Meganol D (Fig 3B). Intriguingly, Algal oil increased the expression of the bacterial LPS co-receptor CD14 (Fig 2B). Antibody cross-linking of CD14 has recently been described to activate MerTK signalling and promote human macrophage clearance of apoptotic cells describing a dual role of CD14 between pro-inflammatory and pro-resolving macrophages [41]. CD36 is another receptor that belongs to the scavenger receptor family involved in phagocytosis, which expression appeared to be increased in presence of Meganol E and Meganol ED (Fig 2B). On macrophages, CD36 is known to contribute to atherosclerosis progression via modified oxLDL phagocytosis and the formation of foam cells [42]. However, CD36 also experts protective functions by enhancing phagocytic ability of macrophages, indicating a context-dependent role for CD36 in the regulation of the initiation and resolution of inflammation [29]. These results support the hypothesis that each of the oils differentially regulates macrophage function, likely *via* the upregulation of distinct SPM pathways.

Two key factors that have been linked with the prevention of vascular inflammation and progression of atherosclerosis are the maintenance macrophage phagocytic capacity and the reduction of macrophage neutral lipid accumulation. In the present study, we demonstrated that the four oils tested increased macrophage phagocytosis and reduced oxLDL uptake in a dose-dependent manner, with Algal oil and Meganol D displaying the most potent responses (Figs 4 and 5). Of note, recent findings demonstrate that RvD1 promotes the resolution of atherosclerotic lesions [22] whereas RvD5$_{n-3 \text{ DPA}}$ protects from the formation of early atherosclerotic lesions. RvE1 also displays potent athero-protective actions *via* the reduction of oxLDL uptake by macrophages [22, 43–45]. In the present study, we found that all four oils increased SPM formation *in vitro* and that both Algal oil and Meganol D upregulated plasma SPM levels

in mice. Notably, Meganol D led to the greatest increases in SPM concentrations both *in vitro* and *in vivo*. This increase in SPM concentrations was linked with an upregulation of macrophage phagocytic ability and decreased uptake of oxLDL. Furthermore, inhibition of two key SPM biosynthetic enzymes, namely ALOX5 and ALOX15, reversed the macrophage directed actions of all four oils.

The strength of the present study is that it provides a side by side comparison of the ability of different oils to regulate both SPM formation and macrophage biology. There are some limitations that should be considered when evaluating the present findings. Our studies with primary human cells employ cells isolated from healthy volunteers. Recent studies indicate that the expression and/or activity of SPM biosynthetic enzymes may be altered in disease [46, 47]. Thus, essential FA supplements may not be efficiently converted to bioactive meditators blunting the effectiveness of this approach in regulating inflammation in patients with chronic inflammatory conditions. Genetic factors, including single nucleotide polymorphism (SNP), in FA and [48] SPM biosynthetic pathways and their receptors will also need to be considered when evaluating the potential utility of omega-3 fatty acid supplements in regulating inflammation [47]. In. this context, recent studies demonstrate that the ability of the EPA-derived SPM RvE1 to reverse hyperinsulinemia and hyperglycemia in obese outbred mice was linked with SNPs identified in its cognate receptor ERV1/ChemR23 [49]. Therefore, future studies we will need to evaluate the efficiency of conversion to bioactive mediators for these supplements in target disease population(s) in order to evaluate their potential therapeutic utility. Another aspect to note is that while in the present studies we investigated the potencies of oils that were delivered in different forms (i.e. ethyl ester vs triglyceride) future systematic studies will need to shed more light into both the influence of fatty acid composition, including their free acid form [50], as well as batch effects in influencing the conversion of fatty acids to SPM. These studies will also need to evaluate the impact of other components found within these oils on regulating both the expression and activity of SPM biosynthetic enzymes. This will help identify components that inhibit the conversion of essential fatty acids to SPM and therefore facilitate the preparation of oils with more effective immunomodulatory actions.

The role of persistent inflammation has long been recognized in the progression of atherosclerosis and other chronic inflammatory conditions, and defects in inflammation resolution have been identified as a key causal factor [24]. SPM exert potent protective actions in the resolution of inflammation, and supplementation with omega-3 enriched oils, which increase SPM concentrations, is linked with reduced vascular inflammation in chronic inflammatory disorders including atherosclerosis. Results from the present study demonstrate that conversion of omega-3 fatty acids and SPM precursors to the bioactive molecules differs between distinct supplements. Thus, supporting the utility of measuring SPM concentrations as potential prognostic markers in determining the efficacy of a given supplement to regulate inflammation. These results also reinforce the need for determining the pharmacodynamics of omega-3 supplements, given the distinct activity observed by each of the supplements tested in regulating key biological actions in the resolution of vascular inflammation.

## Methods

### Omega-3 enriched oils preparation

Four different commercially available omega-3 oils: Algal oil (lot 180322A, vegetarian omega-3 source derived from Algae), and Meganol D (L020218002), Meganol E (L040618171) and Meganol ED (L040618172), fish oil origin omega-3 supplements, were provided by Standard Process Inc. (Palmyra, WI). Meganol D and Meganol E were provided in the Ethyl Ester form and Meganol ED and Algal oil in triglyceride form. Oil volume was adjusted as described

below. Prior to addition to cells or administration *in vivo*, oil suspensions were placed in a water bath sonicator for 10 seconds to disperse micelles and facilitate dissolution [51].

## Human monocyte-derived macrophage preparation and responses

Human peripheral blood mononuclear cells (PBMCs) were isolated from healthy human volunteers purchased from the NHS Blood and Transplant bank and experiments were conducted with Queen Mary Research Ethics Committee approval (QMREC 2014:61) and in accordance with the Helsinki declaration. Informed consent was obtained from all volunteers. Here blood cones were used and PBMCs were isolated by density centrifugation where cells were layered onto Histopaque—1077 (Sigma) and centrifuged for 30 min at 400 x g at room temperature (RT). Macrophages were prepared using published protocols [52] where PBMCs were plated into 10 cm tissue culture plates and incubated at 37°C for 30 min in PBS$^{+/+}$. Non-adherent cells were then removed by adding PBS$^{-/-}$ and washed vigorously. Adherent cells were incubated in RPMI 1640 containing 10% human serum, 1% Penicillin-Streptomycin and 20 ng/mL GM-CSF, and on day 3 media was refreshed after cells were washed with PBS$^{-/-}$. *To establish the ability of each oil to influence SPM biosynthesis in human monocyte-derived macrophages*: On day 7 cells were incubated with the indicated concentrations of oils together with 1 ng/ mL LPS for 24 h. Volumes for each of the oils to be used in the incubations was determined based on the combined amounts of the following SPM precursors 18-HEPE, 17-HDHA, 7-HDHA, 4-HDHA, 17-HDPA, 13-HDPA, 7-HDPA, 15-HETE and 5-HETE, excluding EPA, DPA and DHA parent FAs. Here 200 ng/mL of SPM precursors, which equated to 4.6 μl, 4.8 μl, 5.3 μl and 10.3 μl of Algal Oil, Meganol D, Meganol E and Meganol ED in a final volume of 5 mL, were added. Supernatants were collected and lipid mediators were identified and quantified using LC-MS/MS based lipid mediator profiling. Cells were then harvested, and the expression of phenotypic markers were assessed as detailed below using flow cytometry. *To assess the ability of the test omega-3 supplements to regulate macrophage phagocytosis of bacteria*: Cells obtained as detailed above were then harvested and seeded into 96 well plates at 5 x10$^5$ cells per well and then incubated with the indicated concentrations of oils without the addition of LPS. To each well we added pHrodo labelled *Staphylococcus aureus (S. aureus)* bioparticles and the extent of bacterial phagocytosis was determined after 1 h by measuring fluorescence using a FLUOstar Omega microplate reader (BMG Labtech). For macrophage incubations using inhibitors, cells were exposed 45 min before to ALOX15 inhibitor (25 μM PD146176; Cambridge Bioscience) or ALOX5 inhibitor (10 μM zileuton; Sigma) following addition of each oil at a concentration of 200 pg/ mL of precursors in RPMI 1640 containing 10% human serum, 1% Penicillin-Streptomycin and 20 ng/ mL GM-CSF. *To determine the ability omega-3 supplements to influence neutral lipid accumulation in macrophages*: Cells were prepared and seeded into 96 well plates as for the phagocytosis assay. To measure oxidized-LDL (oxLDL) uptake, cells were incubated with the different concentration of oils for 24 h, media was replaced with serum free RPMI 1640 and cells incubated with fluorescently labelled dil dye oxLDL (dil-oxLDL; 10 μg/ mL, Invitrogen) for 24 h. Subsequently, cells were washed with PBS$^{-/-}$ to remove excess oxLDL and fluorescence (excitation 554nm / emission 571 nm) was measured using a FLUOstar Omega microplate reader (BMG Labtech).

## Flow cytometry

Flow cytometry was used to determine the phenotypic lineage of the monocyte-derived macrophages. Cells were incubated with 1:1 PBS containing 0.02% BSA and FC block for 30 min at 4°C, then for a further 30 min with the following antibodies (diluted 1:100) APC/Cy7 anti-human CD14 (clone 63D3, Biolegend), PE-Cyanine7 anti-human CD32 (clone 6C4,

eBiosciences), anti-human CD80 (clone 2D10, Biolegend), PE/Cy5 anti-CD64 (clone 10.1, Abcam), AF 488 anti-human CD68 (clone Y1/82A, Biolegend), Brilliant Violet (BV) 650, PerCP/Cy5.5 anti-human CD206 (clone 15–2, Biolegend), APC anti-human CD36 (clone 5–271 Biolegend), BV 421 Mouse Anti-Human MerTK (clone: 590H11G1E3; Biolegend) and PE-CF594 anti-human CD163 (clone GHI/61, BD Biosciences). An LSR Fortessa (BD Biosciences) cell analyser was used to perform multiparameter analysis. Staining was analysed using FlowJo (Tree Star Inc., V10). Ortogonal partial least squares-discrimination analysis (oPLS-DA) were performed using SIMCA 14.1 software (Umetrics, Umea, Sweden). The score plot illustrates the systematic clusters among the observations (closer plots presenting higher similarity in the data matrix), and loading plot representation identifies the variables with the best discriminatory power (Variable Importance in Projection (VIP) greater than 1) that were associated with the distinct intervals and contributed to the clusters observed in the score plot.

## Lipid mediator profiling

For omega-3 enriched supplement profiling, 25 μl of each supplement tested (Algal Oil, Meganol D, Meganol E and Meganol ED) was placed in 1 mL of methanol containing deuterium labelled internal standards (500 pg of $d_5$-RvD2, $d_4$-Leukotriene (LT)B$_4$, $d_5$-Lipoxin (LX)A$_4$, $d_4$-Prostaglandin (PG)E$_2$ and $d_8$-5S-Hydroxy-eicosatetraneoic acid; Cayman Chemicals). Cell supernatants were collected and placed in two volumes of methanol containing deuterium labelled internal standards. Mouse plasma was placed in 4 volumes of methanol containing deuterium labelled internal standards. Deuterium-labelled standards were used to aid in identification and quantification of lipid mediators [24]. Proteins were allowed to precipitate by keeping samples at -20°C for 30 min. Following protein precipitation, samples were then extracted using an ExtraHera (Biotage) and ISOLUTE C18 columns (500 mg, 3 mL; Biotage). Products were brought to dryness using a gentle nitrogen stream and TurboVap LV (Biotage), these were then suspended in phase containing methanol and water, 1:1 (vol/vol).

An LC-MS-MS system, comprising of a Qtrap 6500+ or Qtrap 5500 (SCIEX), Shimadzu SIL-20AC autoinjector, LC-20AD binary pump (Shimadzu Corp.) and Agilent C18 Poroshell column (150 mm × 4.6 mm × 2.7 μm) was used to profile lipid mediators. The gradient was initiated at 20:80:0.01 (vol/vol/vol) methanol/water/acetic acid for 0.2 min this was ramped to 50:50:0.01 (vol/vol/vol) over 12 seconds, maintained for 2 min, then ramped to 80:20:0.01 (vol/vol/vol) over 9 min, and maintained for 3.5 min. The ratio was then ramped to 98:2:0.01 (vol/vol/vol) for 5.5 min. The flow rate was kept at 0.5 mL/ min throughout.

Mediator concentrations were determined using multiple reaction monitoring (MRM) using signature parent ion (Q1) and characteristic daughter ion (Q3) pairs. A minimum of six diagnostic ions were used to confirm their identities, in accordance with published criteria [53]. The peak area of the MRM transition and linear calibration curves with an $r^2$ value of 0.98 to 0.99 were used to quantify each of the molecules. Differences in FA substrate and SPM precursor concentrations between the four omega-3 enriched supplements, and differences in plasma lipid mediator concentrations between the different experimental groups, were analysed using Metaboanalyst 3.0 (http://www.metaboanalyst.ca [54]). Data were log transformed and autoscaled before analysis.

## Mouse experiments

*ApoE$^{-/-}$ mice*. All experimental protocols were approved by UK Home Office. Experiments were performed and mice were euthanised following anaesthesia using cervical dislocation in accord with UK Home Office regulations (Guidance on the Operation of Animals, Scientific Procedures Act, 1986) and Laboratory Animal Science Association (LASA) Guidelines

(Guiding Principles on Good Practice for Animal Welfare and Ethical Review Bodies, 3rd Edition, 2015) following methods detailed in a UK Home Office protocol (P998AB295). $ApoE^{-/-}$ mice were obtained from JAX Laboratories (Maine, USA), Mice (male and female) were fed a Western-style diet (LBS Biotech, Cat. # 829100) for 7 weeks from 4 weeks of age and kept in a 12 h light dark cycle. At 9 weeks of age mice were randomized to one of three experimental groups and were given the indicated supplements. A total of 9.2 μg SPM precursors per day, which was equivalent to 20 μL of Meganol D and 27.6 μL of Algal Oil. This was suspended in a final volume of 300 μL and administered to non-anesthetised mice via intraperitoneal injection for a total of 18 days. Mice were then anesthetised using isoflurane and blood was collected via cardiac puncture and lipid mediator profiles were determined as detailed above. Anesthetised mice were then culled via cervical dislocation, aortic arches were collected and stained using oil-red O [55]. Staining intensity was determined using ImageJ and expressed as relative units per $mm^2$.

## Statistical analysis

All results are presented as mean ± SEM. Statistical differences between groups were determined using one-sample t test (for normalized data), Mann Whitney test (for 2 groups), 1-way ANOVA and Dunnett's post hoc test (for multiple groups) using GraphPad Prism 6 software. The investigators were not blinded to group allocation for in vitro experiments whereas blinding was performed for *in vivo* experiments. The criterion used to establish statistical significance was p≤ 0.05. Sample sizes and number of replicates for each experiment were determined on the variability observed in prior experiments [55] and/or pilot experiments.

## Supporting information

**S1 File.**
(DOCX)

## Author Contributions

**Conceptualization:** Jesmond Dalli.

**Data curation:** Agua Sobrino, Mary E. Walker, Romain A. Colas, Jesmond Dalli.

**Formal analysis:** Agua Sobrino, Mary E. Walker, Romain A. Colas, Jesmond Dalli.

**Funding acquisition:** Jesmond Dalli.

**Writing – original draft:** Agua Sobrino.

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
