## [Decision Letter · Decision Letter 0]

2 Oct 2020

PONE-D-20-28326

Pro-Resolving Activities Of Distinct Omega-3 Enriched Oils Are Linked To Their Ability To Regulate Specialized Pro-Resoling Mediators

PLOS ONE

Dear Dr. Dalli,

Thank you for submitting your manuscript to PLOS ONE. After careful consideration, we feel that it has merit but does not fully meet PLOS ONE’s publication criteria as it currently stands. Therefore, we invite you to submit a revised version of the manuscript that addresses the points raised during the review process.

We look forward to receiving your revised manuscript.

Kind regards,

Michael Bader

Academic Editor

PLOS ONE

Journal Requirements:

'This work was supported by funding from the European Research Council (ERC) under the European Union’s Horizon 2020 research and innovation programme (grant no: 677542) and the Barts Charity (grant no: MGU0343) to J.D. J.D. is also supported by a Sir Henry Dale Fellowship jointly funded by the Wellcome Trust and the Royal Society (grant 107613/Z/15/Z). This work was also partly funded by a Grant from Standard Process Inc.'

'The funders had no role in study design, data collection and analysis, decision to publish, or preparation of the manuscript.'

b. Additionally, because some of your funding information pertains to [commercial funding//patents], we ask you to provide an updated Competing Interests statement, declaring all sources of commercial funding.

In your Competing Interests statement, please confirm that your commercial funding does not alter your adherence to PLOS ONE Editorial policies and criteria by including the following statement: "This does not alter our adherence to PLOS ONE policies on sharing data and materials.” as detailed online in our guide for authors  http://journals.plos.org/plosone/s/competing-interests.  If this statement is not true and your adherence to PLOS policies on sharing data and materials is altered, please explain how.

c. Please include the updated Competing Interests Statement and Funding Statement in your cover letter. We will change the online submission form on your behalf.

Reviewers' comments:

Reviewer's Responses to Questions

**Comments to the Author**

1. Is the manuscript technically sound, and do the data support the conclusions?

Reviewer #1: Yes

Reviewer #2: Yes

2. Has the statistical analysis been performed appropriately and rigorously? 

Reviewer #1: Yes

Reviewer #2: Yes

3. Have the authors made all data underlying the findings in their manuscript fully available?

Reviewer #1: Yes

Reviewer #2: Yes

4. Is the manuscript presented in an intelligible fashion and written in standard English?

Reviewer #1: Yes

Reviewer #2: Yes

5. Review Comments to the Author

Reviewer #1: The manuscript by Sobrino et al. demonstrates that administration of differing LC n-3 PUFA-enriched oils results in differential upregulation of specialized pro-resolving mediators (SPMs), despite normalization of the dosing across the supplements. The work was conducted with in vitro and in vivo models and relied on human macrophages from donors and a standard mouse model of APOE-/- mice on a westernized diet. The data are of interest to the field as they highlight differences in differing formulations of marine oils and their impact on SPM levels and associated cellular function. The data appear to be collected in a rigorous manner. A few minor limitations need to be addressed to warrant publication, which include providing additional methodological details and correcting some statements in the intro/discussion with supporting references.

1. Line 77- It is not fair to say that robust biomarkers are not there as the RBC index for EPA/DHA intake is established (PMID: 30837013). In addition, several other lipid pools have also been examined as potential biomarkers of LC n-3 PUFAs (PMID: 26247960). Please acknowledge this.

2. There is no recognition in the text that EPA ethyl esters (Vascepa) are now clinically approved by the United States FDA for lowering CVD risk. Both the introduction and discussion are presented in a manner that there is ambiguity about EPA/DHA for CVD. In fact, the ambiguity has strongly decreased in the past few years. Please present and discuss the following: PMID: 32959713; PMID: 32951855; PMID: 32860032 in either the introduction or discussion. Also, clarify the statements on CVD risk and marine oils.

3. Please fix grammar on line 94.

4. More details are needed in lines 377 and below about the oils. Is the entire composition available to get a sense of “other components” in these oils that may interfere with upregulation of SPMs? Also why was 14-HDHA not one of the SPM precursors listed on line 137?

5. What was the level of variability between batches of oils? For figure 1, are the data representative of 3 separate measurements of the same batch of oil, or were different batches of oil measured? As an example, do 7-HDPA levels for Algal Oil represent 3 batches of oils or 3 measurements of the same batch of oil? This is a concern as one would expect strong variability between batches of each oil tested. At the very least, this should be discussed as a limitation of the study regarding batch-to-batch variation of oils.

6. Supplemental Table 1 should be in the main text.

7. Across data sets, how was the number of independent experiments determined a priori?

8. Line 674 for Figure 3 Legend – it is stated that 200ng of SPM precursors was the concentration used for treatment. Did this concentration of 200ng also include the concentration of the EPA/DPA/DHA in the oils? Perhaps this needs to be clarified as the main text leads the reviewer to think that the 200ng also includes the concentration of parent fatty acids.

9. Are the concentration of Meganol E and Meganol ED used for the phagocytosis studies biologically relevant for concentrations of SPMs achieved in circulation of humans?

10. The title for the Figure 6 legend is confusing and should be clarified.

11. How does the dose of 9.2pg of SPM precursors in the mouse model relate to what is achievable in humans?

12. Discussion - the notion that inhibition of ALOX5 and ALOX15 reversed the effects of SPMs is stated several times and is redundant.

13. Lines 323-334. The Discussion needs some improvement. First, there needs to be more discussion on the fact that other components (besides SPM precursors and EPA/DHA) are likely to vary between the 4 tested supplements. What are those additional components (again, it would be useful to have the composition of these components in a supplemental table as well, if possible) and how do they potentially interfere with the effects of SPM precursors and their parent fatty acids? Second, there needs to be more discussion following lines 325-326 that genetics will also have a strong effect on the response to EPA/DHA (PMID: 29068398) and downstream SPMs (PMID: 32579292). Finally, acknowledge that other forms besides TG and ethyl esters should be tested (i.e. carboxylic acids) in future studies.

Reviewer #2: This is an interesting paper that characterizes different type of omega-3 oils, with regard to their SPM precursor content, their ability to evoke actions on macrophages, and their ability to synthesize SPMs in macrophages and ultimately their protection in a murine model of atherosclerosis. This is well written and well controlled. The experiments are clear and well justified. I have no concerns and I think this is interesting work.

6. PLOS authors have the option to publish the peer review history of their article (what does this mean?). If published, this will include your full peer review and any attached files.

Reviewer #1: No

Reviewer #2: No

---

## [Author Response · Author response to Decision Letter 0]

3 Nov 2020

Reviewer Comments:

Reviewer 1: 

The manuscript by Sobrino et al. demonstrates that administration of differing LC n-3 PUFA-enriched oils results in differential upregulation of specialized pro-resolving mediators (SPMs), despite normalization of the dosing across the supplements. The work was conducted with in vitro and in vivo models and relied on human macrophages from donors and a standard mouse model of APOE-/- mice on a westernized diet. The data are of interest to the field as they highlight differences in differing formulations of marine oils and their impact on SPM levels and associated cellular function. The data appear to be collected in a rigorous manner. A few minor limitations need to be addressed to warrant publication, which include providing additional methodological details and correcting some statements in the intro/discussion with supporting references.

We thank the reviewer for their time in reviewing our manuscript and their interest in the work presented.

1. Line 77- It is not fair to say that robust biomarkers are not there as the RBC index for EPA/DHA intake is established (PMID: 30837013). In addition, several other lipid pools have also been examined as potential biomarkers of LC n-3 PUFAs (PMID: 26247960). Please acknowledge this.

We thank the reviewer for raising this point. We think that there is a misunderstanding here and have clarified this in the revised the text. We also included a reference to the highlighted studies. The revised text is on page 3 lines 92-95 as detailed below:

‘In this context, recent studies have suggested that distinct FA pools such as red blood cells and another plasma FA pools may be useful to evaluate DHA and EPA loading after fatty acid supplementation [17,18].’

2. There is no recognition in the text that EPA ethyl esters (Vascepa) are now clinically approved by the United States FDA for lowering CVD risk. Both the introduction and discussion are presented in a manner that there is ambiguity about EPA/DHA for CVD. In fact, the ambiguity has strongly decreased in the past few years. Please present and discuss the following: PMID: 32959713; PMID: 32951855; PMID: 32860032 in either the introduction or discussion. Also, clarify the statements on CVD risk and marine oils. 

We thank the reviewer for the comment and for the references cited from new studies which reinforced the use of Omega-3 supplements. We are agreed that the ambiguity has strongly decreased in the past few years but divergent results exist and we still consider that it needs to be addressed. In this study we wanted to emphasize more on the role of SPM formation after supplementation of their parent FAs to achieve such benefits. We have revised the manuscript to include the studies indicated. The revised text can be found on page lines 73-82 as detailed below:

‘Recent clinical studies have further underscored the utility of omega-3 fatty acids in the treatment of CVD. Whereby, in the REDUCE-IT trial Icosapent ethyl (IPE), a highly purified form of eicosapentaenoic acid ethyl ester, added to a statin reduces initial cardiovascular (CV) events by 25% and total CV events by 32 [11] The recently concluded EVAPORATE study demonstrated significant administration of IPE together with statin led to increased regression of low-attenuation plaque volume on multidetector computed tomography compared with placebo over 18 months [12] Furthermore, recent meta-analysis and meta-regression of interventional trials provided further support for the use of enriched omega -3 FAs, EPA and docosahexaenoic acid (DHA) as an effective lifestyle strategy for CVD prevention [13].’ 

3. Please fix grammar on line 94.

Thank you, this is revised.

4. More details are needed in lines 377 and below about the oils. Is the entire composition available to get a sense of “other components” in these oils that may interfere with upregulation of SPMs? Also why was 14-HDHA not one of the SPM precursors listed on line 137?

Thank you for this intriguing question. Unfortunately we do not have the entire composition of the oils. As the reviewer may appreciate give that these oils are derived from natural sources their complete composition is not routinely determined, an aspect that has to-date complicated the interpretation of findings made with such preparations. Our study attempts to address this knowledge gap by determining the monohydroxy fatty acid composition of these oils in addition to the routinely determined essential fatty acid content. The influence of such unknown components on SPM production is indeed a very interesting question and one that will be subject to future investigations since addressing this question will help in the preparation of oils that are likely to have potent immunoregulatory actions. 

The reason why 14-HDHA was not included as an SPM precursor is that in the maresin biosynthetic pathway, it is the 14-hydroperoxide that serves as a precursor to SPM biosynthesis, and not 14-HDHA, which is instead a pathway marker for this SPM family (PMID: 19103881; PMID: 2350471).

5. What was the level of variability between batches of oils? For figure 1, are the data representative of 3 separate measurements of the same batch of oil, or were different batches of oil measured? As an example, do 7-HDPA levels for Algal Oil represent 3 batches of oils or 3 measurements of the same batch of oil? This is a concern as one would expect strong variability between batches of each oil tested. At the very least, this should be discussed as a limitation of the study regarding batch-to-batch variation of oils.

We thank the reviewer for pointing this out. At the time of the study was performed we only had access to one batch of the oils. While we agree that determining batch to batch variation in SPM precursor concentrations and activity of the oils, we believe this is outside the scope of the present manuscript. Indeed, the scope of the present study is to evaluate the efficacy of conversion of SPM precursors to the bioactive autacoids with distinct oil preparations and the relationship this has to the regulation of immune cell response. In the revised manuscript we address the need for future studies to determine the batch to batch effects on SPM formation. The revised text is on page 12 of the manuscript and detailed below:

‘Another aspect to note is that while in the present studies we investigated the potencies of oils that were delivered in different forms (i.e. ethyl ester vs triglyceride) future systematic studies will need to shed more light into both the influence of fatty acid composition, including their free acid form [50], as well as batch effects in influencing the conversion of fatty acids to SPM.

6. Supplemental Table 1 should be in the main text.

Thank you, Supplemental table 1 has been moved to the main text as Table 1.

7. Across data sets, how was the number of independent experiments determined a priori?

This was determined using pilot data. This is included in the revised text on page 16 lines 519-521 as detailed below:

‘Sample sizes and number of replicates for each experiment were determined on the variability observed in prior experiments [55] and/or pilot experiments.’

8. Line 674 for Figure 3 Legend – it is stated that 200ng of SPM precursors was the concentration used for treatment. Did this concentration of 200ng also include the concentration of the EPA/DPA/DHA in the oils? Perhaps this needs to be clarified as the main text leads the reviewer to think that the 200ng also includes the concentration of parent fatty acids.

We thank the reviewer for bringing this misimpression. 200ng of SPM precursors was the concentration used for treatment excluding of the EPA/DPA/DHA. We have clarified this point in the methods on page 13 lines 412-413.

9. Are the concentration of Meganol E and Meganol ED used for the phagocytosis studies biologically relevant for concentrations of SPMs achieved in circulation of humans?

Thank you for the insightful question. The concentration used of all oils in the present studies are relevant to plasma concentrations of SPM precursors that are achieved after fish oil supplementation in humans, where concentrations as high as 2200 pg/mL of these precursors can be achieved after a bolus administration. This is discussed on page lines 157-159 of the revised manuscript as detailed below:

‘This concentration was selected since it is relevant to plasma levels of these molecules achieved in humans after supplementation with fish oils [21]. 

10. The title for the Figure 6 legend is confusing and should be clarified.

This is revised as detailed below:

‘Figure 6. Inhibition of ALOX5 and ALOX15 reduced the ability of omega-3 enriched oil to upregulate to macrophage phagocytosis increase of S. aureus bioparticles.’

11. How does the dose of 9.2pg of SPM precursors in the mouse model relate to what is achievable in humans?

We thank the reviewer for this intriguing question. Given the physiological and metabolic differences between mice and humans this aspect would need to be determined empirically in humans. Nonetheless, results from published studies using a different omega-3 supplement suggest that potentially lower doses of SPM precursors may be required to achieve comparable plasma SPM concentrations in both human healthy volunteers and patients (PMID: 31829100) with peripheral artery disease (PMID: 32696697)

12. Discussion - the notion that inhibition of ALOX5 and ALOX15 reversed the effects of SPMs is stated several times and is redundant.

We thank the reviewer for the comment we have revised the text to remove repetition

13. Lines 323-334. The Discussion needs some improvement. First, there needs to be more discussion on the fact that other components (besides SPM precursors and EPA/DHA) are likely to vary between the 4 tested supplements. What are those additional components (again, it would be useful to have the composition of these components in a supplemental table as well, if possible) and how do they potentially interfere with the effects of SPM precursors and their parent fatty acids? Second, there needs to be more discussion following lines 325-326 that genetics will also have a strong effect on the response to EPA/DHA (PMID: 29068398) and downstream SPMs (PMID: 32579292). Finally, acknowledge that other forms besides TG and ethyl esters should be tested (i.e. carboxylic acids) in future studies. 

Thank you for these suggestions, we have included the above points in the revised discussion on pages 11-12 lines 351-357 and 360-369 as detailed below:

‘Genetic factors, including single nucleotide polymorphism (SNP), in FA and [48] SPM biosynthetic pathways and their receptors will also need to be considered when evaluating the potential utility of omega-3 fatty acid supplements in regulating inflammation [47]. In. this context, recent studies demonstrate that the ability of the EPA-derived SPM RvE1 to reverse hyperinsulinemia and hyperglycemia in obese outbred was linked with SNPs identified in its cognate receptor ERV1/ChemR23 [49].

‘Another aspect to note is that while in the present studies we investigated the potencies of oils that were delivered in different forms (i.e. ethyl ester vs triglyceride) future systematic studies will need to shed more light into both the influence of fatty acid composition, including their free acid form [50], as well as batch effects in influencing the conversion of fatty acids to SPM. Future studies will also need to evaluate the impact of other components found within these oils on regulating both the expression and activity of SPM biosynthetic enzymes. This will help identify components that inhibit the conversion of essential fatty acids to SPM and therefore facility the preparation of oils with more effective immunomodulatory actions.’

Reviewer #2: 

This is an interesting paper that characterizes different type of omega-3 oils, with regard to their SPM precursor content, their ability to evoke actions on macrophages, and their ability to synthesize SPMs in macrophages and ultimately their protection in a murine model of atherosclerosis. This is well written and well controlled. The experiments are clear and well justified. I have no concerns and I think this is interesting work.

We thank the reviewer for their encouraging comments and their time to review our manuscript.

---

## [Editor Report · Decision Letter 1]

5 Nov 2020

Pro-Resolving Activities Of Distinct Omega-3 Enriched Oils Are Linked To Their Ability To Upregulate Specialized Pro-Resoling Mediators

PONE-D-20-28326R1

Dear Dr. Dalli,

We’re pleased to inform you that your manuscript has been judged scientifically suitable for publication and will be formally accepted for publication once it meets all outstanding technical requirements.

Kind regards,

Michael Bader

Academic Editor

PLOS ONE
---

## [Editor Report · Acceptance letter]

7 Dec 2020

PONE-D-20-28326R1 

Protective Activities Of Distinct Omega-3 Enriched Oils Are Linked To Their Ability To Upregulate Specialized Pro-Resolving Mediators 

Dear Dr. Dalli:

I'm pleased to inform you that your manuscript has been deemed suitable for publication in PLOS ONE. Congratulations! Your manuscript is now with our production department. 

Kind regards, 

on behalf of

Prof. Michael Bader 

Academic Editor

PLOS ONE